# Unveiling the significance of women's role in health-seeking behavior during suspected malaria fever in risk populations of Nepal: Mixed methods cross-sectional study

**Ashok Kumar Paudel** [1]*, **Muni Raj Chhetri**[2], **Nibha Rani Pandey**[3], **Prem Prasad Panta**[4]

**1** Department of Research and Development, National Open College, Pokhara University, Lekhnath, Nepal,
**2** Department of Public Health, National Open College, Pokhara University, Lekhnath, Nepal, **3** Ram Krishna Dharmartha Foundation (RKDF) University, Ranchi, India, **4** KIST Medical College & TH, TU, Lalitpur, Nepal

* ashokkpaudel@gmail.com

**Data Availability Statement:** All data underlying the findings described in this manuscript will be

## Abstract

Malaria remains a substantial global health challenge, causing preventable illnesses and fatalities. In Nepal, the government has ambitiously targeted achieving malaria-free status by 2025. This study aims to assess the impact of women's roles on health-seeking behavior during suspected malaria fever in the high-risk area of Kanchanpur district, Nepal. This is a cross-sectional analytical design with a mixed-method approach, the research focused on Kanchanpur district, selected from 20 high-risk malaria districts of Nepal. Belauri Municipality within Kanchanpur, identified for its concentration of high and moderate-risk wards, was the specific study area. A random selection process identified 387 households for a comprehensive survey. Face-to-face interviews with household heads were conducted after obtaining written informed consent and ethical approval from the Nepal Health Research Council (March 3, 2023/Ref no.-2041). Data analysis, employing statistical measures such as percentages, frequency, mean, and the Chi-square test, was performed using SPSS version 20. Cultural beliefs regarding women's use of bed nets during menstruation significantly predicted health-seeking behavior (p-value < 0.05). Those endorsing bed net use during menstruation were nearly twice as likely to choose modern health facilities (COR = 1.975, 95% C.I. = 1.134 to 3.439, p = 0.016). Women's involvement in malaria treatment decisions strongly correlated with health-seeking behavior (p-value = 0.001). However, women participating in household decisions for suspected malaria treatment were less likely to choose modern health facilities (COR = 0.327, 95% CI = 0.171–0.627, p = 0.001) compared to those without such a role. The study underscores the complex influence of cultural beliefs and women's decision-making roles on health-seeking behavior. Recognizing and comprehending these factors are vital for crafting effective malaria interventions that align with cultural contexts and consider the nuanced roles of women in health-related decisions.

freely available to other researchers. Data file has been uploaded as supplementary information.

**Funding:** The authors received no specific funding for this work.

**Competing interests:** The authors have declared that no competing interests exist.

## Introduction

Bio-medically, malaria is a well-defined disease in which parasites of the genus *Plasmodium* are transmitted to humans through the bite of female mosquitoes belonging to the genus Anopheles [1,2]. The inception of the World Health Organization's Rollback Malaria program in 1998 aimed to achieve a 50% reduction in malaria cases by 2010, while the Global Technical Strategy for Malaria 2016–2030 strives for a 90% decrease in malaria's burden by 2030 [3,4]. Globally, although there was a positive decline in malaria incidence rates from 81 cases per 1,000 people at risk in 2000 to 57 cases per 1,000 in 2019, the COVID-19 pandemic significantly impacted progress. In 2022, the number of global malaria cases surged to 249 million, exceeding pre-pandemic levels. This rise indicates a concerning increase in incidence rates, despite only a slight increase observed in 2020 [5].

A study undertaken in Nepal in 2018 revealed that out of the country's 77 districts, 67 are situated in areas categorized as high, moderate, or low risk for malaria. Moreover, around 43.26% of the entire population resides in these at-risk regions. The study's findings pinpointed that within the new federal structure, 202 wards spanning 20 districts bear a high or moderate malaria risk. Specifically, in Nepal, roughly 3.96% of the total populace inhabits malaria endemic areas, comprising 0.22 million individuals in high-risk zones (49 wards), 0.93 million in moderate-risk zones (253 wards), and 11.34 million in low-risk zones (2,543 wards) [6]. In Nepal, confirmed malaria cases increased from 491 in Fiscal Year 2022/23 to 533 in Fiscal Year 2023/24, predominantly due to imported cases (95.6%). The incidence of *Plasmodium falciparum* infections also escalated, representing over 28.33% of all cases in Fiscal Year 2023/24. Among Nepal's total confirmed malaria cases, Kanchanpur district reported 28 cases in Fiscal Year 2023/2024 [7].

Previous studies have reported that health and illness are defined, labeled, evaluated, and acted upon in the context of social and cultural aspects of human life [8–10]. Any activities under taken by individuals, who perceive them to have a health problem or to be ill for the purpose of finding appropriate remedy is health seeking behavior [11–13]. It is influenced by a large number of individual and social factors; is a complex outcome of many factors operating at individual, family and community level [14]. Health care seeking behavior and the process of illness responses and the decision to engage with a particular type of health care are influenced by a variety of socio-cultural factors [15]. Therefore, this mixed-method cross-sectional analytical study was conducted to assess the women's role affecting health-seeking behavior during suspected malaria fever in the high-risk area of Kanchanpur district, Nepal.

## Materials and methods

### Study design and setting

In this study, a cross-sectional analytical design with a mixed-method approach was implemented from March 15th to May 15th, 2023. Among the 20 districts in Nepal with malaria risk, Kailali and Kanchanpur districts stood out with higher numbers of high and moderate-risk wards (58 in Kailali and 43 in Kanchanpur). Using a simple random sampling method, Kanchanpur district was chosen. Within Kanchanpur, Belauri Municipality, housing the highest numbers of high and moderate-risk wards for malaria, was selected. Ward number 1, the sole high-risk ward in Belauri, was chosen for the study. Additionally, 50% of the 8 moderate-risk wards were randomly selected, totaling 5 wards (1, 2, 3, 4, and 6). Comprehensive household details for Belauri Municipality and the selected wards were prepared.

## Study population

As per the Municipal profile of Belauri Municipality in 2018, the selected wards (1, 2, 3, 4, and 6) comprised a study population of 4,637 households with a total population of 24,852. All households of selected wards were the study population for household survey.

## Sample size and sampling process

The sampling unit for this study was households from the selected wards. Using the formula (n) = N / (1+N∗e$^2$), where N is the population under study (4,637), n is the sample size, e is the level of precision (5%) with 95% confidence, and P is 0.5, the calculated sample size (n) was 368. Considering a non-response rate of 5% (19 households), the total sample size became 387. The Probability Proportional to Size (PPS) method determined the sample size for each selected ward, with the Simple Random Sampling (SRS) method identifying the desired number of households from each ward.

## Data collection tool and technique

A structured interview questionnaire was employed for the household survey, with heads of households serving as respondents. Individuals diagnosed with malaria infection during the last one-year period were chosen as respondents for In-depth Interviews (IDIs). Participants with malaria infection diagnosed by local health facilities were deliberately selected based on the records of those facilities. Two IDIs were conducted in each ward, totaling 10 IDIs to conclude the study. Health care providers and managers, teachers, social workers, community leaders, Female Community Health Volunteers (FCHVs), and traditional healers (TH) were respondents for Key Informant Interviews (KIIs). A total of 10 KIIs were conducted to complete the study.

Men and women above 18 years old, residing within selected wards, participated in Focus Group Discussions (FGDs). Eight to ten people meeting the study population criteria were purposively selected for each FGD. Two FGDs were conducted in each ward, resulting in a total of 10 FGDs among different social and ethnic groups.

Open-ended guiding questions were developed and presented to the respondents for the IDI and KII methods of data collection. A list of open questions or issues was developed to gather information through FGD techniques. For completeness, the collected data underwent editing, review, and checks.

## Data analysis

Statistical Package for Social Sciences (SPSS) version 20 was employed to analyze the survey data. For the data generated from the household survey, percent distribution was used for descriptive analysis, and comparisons were made using the Chi-square test to assess associations between dependent and independent variables.

Qualitative data from In-depth Interviews (IDIs), Key Informant Interviews (KIIs), and Focus Group Discussions (FGDs) were analyzed manually. Initially, the qualitative data was organized using a thematic analysis approach. Key issues and themes were identified, and the responses to questions within these themes were grouped and summarized in a data analysis framework. Quotations that illustrated the views of the majority of participants or those in contradiction with the majority were extracted. These issues were then summarized by wards and health facility levels and finally integrated into the relevant sections. The data was further summarized by using all the original texts, listing all conceptual categories and patterns, and

placing relevant information under these conceptual categories. Relationships were identified between the categories.

### Ethical approval and informed consent

Before data collection, the study's purpose was explained to the respondents, and written informed consent was obtained. Ethical Approval was obtained from Nepal Health Research Council/Nepal (March 3, 2023/Ref no.-2041).

## Results

### Background characteristics of household survey respondents

Table 1 is about the socio-demographic features of household survey respondents. Study reveals that the distribution of ages is skewed towards the younger age ranges, with the more than half (53.5%) of respondents falling in the 30–50 age range. There is a smaller proportion (22.0%) of respondents fall within the 20–30 age and 13.4% in 50–60 age group, and an even smaller proportion (11.1%) among 60 and above age group. The distribution of sex is skewed towards males, with 70.5% of the respondents being male and 29.5% being female. Study also reveals that marital status was heavily weighted towards respondents who are currently married (87.6%) followed by widowed (7.2%), single/never married (3.1%) and divorced (2.1%). The overwhelming majority of respondents were Hindu (89.4%). There were also smaller proportion of respondents who identify as Christian (7%), Buddhist (3.1%) and Muslim (0.5%).

**Table 1. Socio-demographic features of survey respondents (N = 387).**

| Variables | Category | Frequency | Percent |
|---|---|---|---|
| Age (in years) | 20–30 | 85 | 22.0 |
| | 30–40 | 110 | 28.4 |
| | 40–50 | 97 | 25.1 |
| | 50–60 | 52 | 13.4 |
| | 60 and above | 43 | 11.1 |
| | Mean = 41.33 yrs; SD = 12.7 yrs. | | |
| Sex | Male | 273 | 70.5 |
| | Female | 114 | 29.5 |
| Marital status | Currently Married | 339 | 87.6 |
| | Divorced | 8 | 2.1 |
| | Single/Never married | 12 | 3.1 |
| | Widow/Widower | 28 | 7.2 |
| Religion | Hindu | 346 | 89.4 |
| | Buddhist | 12 | 3.1 |
| | Muslim | 2 | 0.5 |
| | Christian | 27 | 7.0 |
| Caste/ethnicity | Bhrahmin | 14 | 3.6 |
| | Chhetri | 109 | 28.2 |
| | Tamang | 12 | 3.1 |
| | Dalit | 127 | 32.8 |
| | Janajati | 53 | 13.7 |
| | Tharu | 72 | 18.6 |
| Family type | Nuclear | 267 | 69.0 |
| | Joint | 120 | 31.0 |

**Table 2. Availability and use of bet nets to protect from mosquito biting.**

| Characteristics | Category | Frequency | Percent |
|---|---|---|---|
| Availability of bet nets (n = 387) | Yes | 350 | 90.4 |
| | No | 37 | 9.6 |
| Types and numbers of bed nets (n = 350) | Insecticide treated bed nets | 329 | 94.0 |
| | Ordinary bed nets | 21 | 6.0 |
| Everybody of the house sleep under a bed net? (n = 350) | Yes | 274 | 78.3 |
| | No | 76 | 21.7 |
| Reasons for everybody of the house does not sleep under a net (n = 76) | Do not have space/room for bed nets | 31 | 40.8 |
| | Do not like to use bet nets | 25 | 32.9 |
| | Do not have sufficient no. of bet nets | 20 | 26.3 |

The largest number of respondents identify as Dalit (32.8%) followed by Chhetri (28.2%), Tharu (18.6%), Janajati (13.7%), Tamang (3.1%), and Bhrahmin (3.6%). Sixty-nine percent of the respondents belong to nuclear families and 31% coming from joint families.

## Availability and use of bed nets in households

Overall data of Table 2 suggests that while the majority of respondents have bed nets (90.4%) available to them and use them to protect against mosquito bites, there is still a significant number of people who do not sleep under bed nets due to various factors. This may be an area of concern, as sleeping under a bed net can significantly reduce the risk of being bitten by mosquitoes and contracting malaria.

## Cultural belief and use of bed nets during menstruation

Table 3 reveals that 72.6% of the household survey respondents believe that women can use bed nets during menstruation, while 27.4% believe that women cannot use bed nets during menstruation. It is important to note that there is no scientific basis for the belief that women cannot use bed nets during menstruation. Bed nets are an effective means of protecting against mosquito bites and can be used by anyone, regardless of their menstrual status. In fact, using a bed net can help to protect women from mosquito bites and reduce their risk of contracting malaria or other mosquito-borne diseases.

## Theme: Cultural belief about using bed nets during menstruation

FGD and KII results also shows the complexity of cultural, and familial beliefs regarding the use of bed nets during menstruation, highlighting the diverse perspectives and changes occurring within different communities.

> " Didn't know any religious belief in my family that said we should not sleep under bed nets during menstruation"(Tharu female 36 yrs/FGD)

**Table 3. Belief about women using bed nets during menstruation (N = 350).**

| Characteristics | Frequency | Percent |
|---|---|---|
| Women can use bed nets during menstruation | 254 | 72.6% |
| Women cannot use bed nets during menstruation | 96 | 27.4% |

**Table 4. Women's role in household decisions for seeking health care during suspected malaria fever.**

| Characteristics | Category | Frequency | Percent |
|---|---|---|---|
| Do women have role in household decisions for seeking care during malaria? (n = 387) | Yes | 342 | 88.4 |
| | No | 45 | 11.6 |
| Women's role in household decisions for seeking care during malaria? (n = 342) | Suggesting for type of treatment | 182 | 53.2 |
| | Making decision for in choice of treatment | 62 | 18.1 |
| | Suggest for financial aspect of treatment | 38 | 11.1 |
| | Making decision for cost of treatment | 30 | 8.8 |
| | Other supportive roles | 30 | 8.8 |

*" My mother in law do not allow to sleep under bed net during menstruation but my husband is supportive so I sleep under bed nets during menstruation" (Female, literate, 32 yrs/FGD)*

*"In past, restriction to use bed nets were more prevalent especially among Brahmin/chhetri groups. But these days, because of education & change in society few of the family do restriction to use bed nets during menstruation" (FCHV, 42 yrs/KII)*

*"My senior family members don't restrict me but if I use bed nets during menstruation they feel unhappy that I have noticed" (Female, illiterate 34 yrs/ FGD)*

### Role of women in household decisions for seeking health care during suspected malaria fever

Table 4 presents the results of a survey on the involvement of women in household decisions about the treatment of malaria. According to the Table 4, 88.4% of the people surveyed reported that women have a role in household decisions about the treatment of malaria, while 11.6% reported that women do not have a role in these decisions. According to tables 4.20, among those who reported that women have a role, the most common roles cited were suggesting the type of treatment (53.2%), making the decision about the choice of treatment (18.1%), suggesting the financial aspects of treatment (11.1%), making the decision about the cost of treatment (8.8%), and other supportive roles (8.8%). These results suggest that a significant proportion of the people surveyed reported that women have an important role to play in household decisions about the treatment of malaria, and that the specific roles played by women may vary.

### Theme: Women's involvement in household decisions for treatment of malaria

FGD and IDI results also shows that the women involvement in decision of suspected malaria treatment is influenced also by belief, family types and roles of husbands & seniors of family.

*"My family don't belief in traditional healers, so in case of suspected malaria fever, me and my husband jointly make decisions for treatment. We prefer local public health facility because it is closer and treatment are in free of cost" (Female from nuclear family, 24 yrs/ FGD)*

*"My self-made decision to go hospital during malaria fever. My family also supported to my decision" (Female, literate from joint family, 32 yrs/IDI)*

*"I have 4 yrs of child. Usually my mother in law suggests me about where to seek treatment for any illness for my son" (Female from joint family, 32 yrs/FGD)*

**Table 5. Association between belief about women using bed nets during menstruation and health seeking behavior (N = 350).**

| Variables | HR/SC (%) | Public HF (%) | Pvt. HF (%) | TH (%) | Total (%) | Chi square | P value* |
|---|---|---|---|---|---|---|---|
| Women can use bed nets during menstruation | 11.8 | 53.5 | 29.9 | 4.7 | 72.6 | 8.630 | 0.03 |
| Women cannot use bed nets during menstruation | 24.0 | 50.0 | 21.9 | 4.2 | 27.4 | | |

* P-value significance at <0.05.

> *"My husband and me make decisions about where to seek treatment for any illness. For minor illness we prefer local public health facility and hospital for serious conditions" (Female from nuclear family, 38 yrs/FGD)*

## Bivariate analysis of the factors associated with health seeking behavior

Chi-square analysis was performed to identify the association between dependent and independent variables. The level of significance has been set to 0.05 or 5% to measure the association between the variables. For this purpose, dependent variables have been grouped as- Home remedy/self-care (HR/SC), Public health facility (HF), Private health facility (HF), and Traditional healer (TH).

Table 5 presents the results of a study on the association between people's belief about whether women can use bed nets during menstruation and their health-seeking behavior for malaria. The study found that when people believe that women can use bed nets during menstruation, the most common health-seeking behavior was going to a government health facility (53.5%), followed by self-administration of drugs (29.9%), and private health facility (4.7%). When people believe that women cannot use bed nets during menstruation, the most common health-seeking behavior was self-administration of drugs (24.0%), followed by going to a government health facility (50.0%), and private health facility (21.9%). The Chi-square test of association shows that belief about whether women can use bed nets during menstruation is a statistically significant predictor of health-seeking behavior (p-value <0.05).

## Multivariate analysis of the factors associated with health seeking behavior

Multivariate analysis for association between belief and health seeking behavior, the logistic regression model demonstrated that individuals who believed women could use bed nets during menstruation were nearly twice as likely to choose modern health facilities (COR = 1.975, 95% C.I. = 1.134 to 3.439, p = 0.016) (Table 6).

Table 7 demonstrates an association between women's involvement in household decisions regarding treatment for suspected malaria fever and health-seeking behavior. Women's involvement in malaria treatment decisions was strongly associated with health-seeking behavior (p value of 0.001). Result reflected that the women's involvement in household decisions for treatment of suspected malaria was less likelihood of choosing modern health facilities

**Table 6. Multivariate analysis of belief on using bed nets during menstruation and health seeking behavior.**

| Variables | Categories | COR | 95% C.I. for OR | | P value |
|---|---|---|---|---|---|
| | | | Lower | Upper | |
| Women can use bed nets during menstruation | Yes | 1.975 | 1.134 | 3.439 | .016 |
| | No | 1 | | | |

**Table 7. Multivariate analysis of women involvement in HHs decisions and health seeking behavior.**

| Variables | Categories | COR | 95% C.I. for OR | | P value |
|---|---|---|---|---|---|
| | | | **Lower** | **Upper** | |
| Women have role in household decisions for treatment of suspected malaria fever | Yes | .327 | .171 | .627 | .001 |
| | No | 1 | | | |

(COR = 0.327, 95% CI = 0.171–0.627, p = 0.001) compared to women without role in such decisions.

## Discussion

The focus of malaria control and prevention efforts is to reduce mortality and infection rates through effective case management and vector control measures, such as promoting appropriate health-seeking behavior, the use of insecticide-treated bed nets, and mosquito elimination [16,17]. Lack of widespread adoption of seeking correct health care from appropriate health facilities and using preventive measures among the affected population is the key obstacle in controlling malaria, making community involvement in malaria prevention essential for successful control programs. It has been understood that health-seeking behavior itself is a complex and dynamic process and individuals don't seek only one source of health care during illness. In this context, understanding the gender role in health care-seeking behavior during malaria fever allows a better appreciation of how and why individuals visit or seek health care with both formal and informal health institutions during malaria fever [18]. Therefore, this study has been conducted to understand the women's involvement affecting health-seeking behavior during suspected malaria fever in high-risk areas of Nepal. The findings of this study will be helpful in strengthening malaria control programs and supporting policymakers, planners, managers, learners, and other interested persons involved in malaria control and elimination activities.

In the current study, 90.4% of households had bed nets, with 85% of these bed nets treated with insecticide (ITN), and 78.3% of the households used them for all household members. However, low usage was still evident despite high ownership. The main reasons for not using the bed nets for all members were lack of space (40.8%), discomfort (32.9%), and an insufficient number of bed nets (26.3%). Concerning the high availability and low use of insecticidal bed nets, previous studies have also reported similar findings to the current study, with 96.8% of households possessing ITNs in South Cameroon [19] and 89.6% of households owning ITNs in Ghana, but only 68.4% utilizing them the night before the study [20]. Despite high ownership, low usage of ITNs has also been reported in previous studies [21–23]. Previous studies have revealed that the inability to mount the ITN due to a lack of an appropriate location in the household, discomfort caused by increased warmth when sleeping under the net, and insufficient number of ITNs to accommodate every household member were major reasons for not using the ITNs [24,25] and these findings are compatible with the current study.

The current study also assessed cultural beliefs about bed net use during menstruation, revealing a significant association with health-seeking behavior (p-value <0.05). Individuals who believe that women can use bed nets are nearly twice as likely to choose modern health facilities (COR = 1.975, 95% C.I. = 1.134 to 3.439, p = 0.016). This result indicates that cultural beliefs about bed net use during menstruation significantly influence healthcare seeking behavior. Specifically, individuals who believe that women can use bed nets during menstruation are almost twice as likely to opt for modern health facilities when seeking

healthcare. The statistical analysis supports this finding with a p-value of 0.016, which is less than the commonly used threshold of 0.05, indicating that the association is statistically significant. Previous studies did not report findings related to cultural beliefs regarding restrictions or permissions on using mosquito bed nets during menstruation. However, some studies have noted similar beliefs about the use of bed nets to prevent mosquito bites. For instance, perceptions, social practices, and beliefs regarding Long-Lasting Insecticidal Nets (LLINs) contribute to the nonuse of LLINs in Madagascar [26]. In Rwanda, among 870 pregnant women, 57.9% (95%CI: 54.6–61.1) used mosquito bed nets, and 16.7% did not use bed nets among those owning bed nets [27]. However, a higher percentage of females used Insecticide-Treated Nets (ITNs) compared to males (57.2% vs. 48.8%) in Kano State, Nigeria [28] and inequalities were reported in the use of insecticide-treated nets by pregnant women in Ghana between 2011 and 2017 [29].

The current study demonstrated that women's involvement in malaria treatment decisions was strongly associated with health-seeking behavior (p = 0.001). The results also indicated that women's involvement in household decisions for the treatment of suspected malaria was associated with a lower likelihood of choosing modern health facilities (COR = 0.327, 95% CI: 0.171–0.627, p = 0.001) compared to women without a role in such decisions. It is important to consider the role of gender in health-seeking behavior during malaria. In this context, a previous study from Myanmar [30] found that family decision-making plays a significant role in the treatment of malaria, as established by multivariable logistic regression analysis. Other studies highlighted the influence of women's access to resources and decision-making power on health-seeking behavior, particularly for febrile illnesses in children [31,32]. Gender-based barriers at the household level have a profound impact on health-seeking behavior. Younger married women were found to delay seeking healthcare the most, as they often had to negotiate with male family members, including husbands and mothers-in-law, who hold the majority of decision-making power. Women who lack financial support from male relatives, or who disagree with male family members about appropriate treatment-seeking, may face difficulties in accessing healthcare for children with malaria [31]. The findings of the current study are consistent with some earlier studies.

The outcomes of the present investigation, carried out in the Kanchanpur district of Nepal, deviate from earlier studies due to variations in research design, participant characteristics, and sampling methods. This study utilized a mixed-method design, encompassing a cross-sectional survey and qualitative components, to collect information on health-seeking behavior amid suspected malaria fever. While the mixed-method approach offers a nuanced comprehension, it's crucial to recognize that causation cannot be established through this study design. Furthermore, limitations such as recall bias and self-reported data susceptible to social desirability bias need to be taken into account. Future research could benefit from diverse study designs and methods to enhance our understanding of the factors influencing health-seeking behavior during suspected malaria fever.

## Conclusions

Cultural belief about whether women can use bed nets during menstruation has impact on health-seeking behavior so addressing and understanding these beliefs can be crucial in malaria prevention and control. Women's involvement in household decisions for the treatment of suspected malaria and the likelihood of choosing modern health facilities. Women without a role in such decisions are less likely to opt for modern health facilities. This highlights the impact of gender dynamics and decision-making power on healthcare choices.

## Supporting information

**S1 Data.**

(SAV)

## Acknowledgments

We acknowledge and appreciate respondents for their valuable time and kind cooperation.

## Author Contributions

**Conceptualization:** Ashok Kumar Paudel.

**Data curation:** Muni Raj Chhetri, Prem Prasad Panta.

**Formal analysis:** Ashok Kumar Paudel, Prem Prasad Panta.

**Funding acquisition:** Muni Raj Chhetri.

**Investigation:** Prem Prasad Panta.

**Methodology:** Ashok Kumar Paudel.

**Project administration:** Muni Raj Chhetri.

**Supervision:** Prem Prasad Panta.

**Validation:** Ashok Kumar Paudel, Muni Raj Chhetri, Nibha Rani Pandey, Prem Prasad Panta.

**Writing – original draft:** Ashok Kumar Paudel, Nibha Rani Pandey.

**Writing – review & editing:** Ashok Kumar Paudel, Muni Raj Chhetri, Nibha Rani Pandey, Prem Prasad Panta.

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
