## [Decision Letter · Decision Letter 0]

27 Mar 2024

PGPH-D-24-00163

Unveiling the significance of women's role in health-seeking behavior during suspected malaria fever in risk populations of Nepal: Mixed methods cross-sectional study

Dear Dr. Paudel,

Thank you for submitting your manuscript to PLOS Global Public Health. After careful consideration, we feel that it has merit but does not fully meet PLOS Global Public Health’s publication criteria as it currently stands. Therefore, we invite you to submit a revised version of the manuscript that addresses the points raised during the review process.

Please note that we have only been able to secure a single reviewer to assess your manuscript. We are issuing a decision on your manuscript at this point to prevent further delays in the evaluation of your manuscript. Please be aware that the editor who handles your revised manuscript might find it necessary to invite additional reviewers to assess this work once the revised manuscript is submitted. However, we will aim to proceed on the basis of this single review if possible. The reviewer has raised multiple major concerns, specifically they recommend that you update your references, and use the most recent available data. Could you please carefully revise the manuscript to address all comments raised?

We look forward to receiving your revised manuscript.

Kind regards,

Johanna Pruller, Ph.D.

Staff Editor

Journal Requirements:

Additional Editor Comments (if provided):

Reviewers' comments:

Reviewer's Responses to Questions

**Comments to the Author**

1. Does this manuscript meet PLOS Global Public Health’s publication criteria? Is the manuscript technically sound, and do the data support the conclusions? The manuscript must describe methodologically and ethically rigorous research with conclusions that are appropriately drawn based on the data presented.

Reviewer #1: Yes

2. Has the statistical analysis been performed appropriately and rigorously?

Reviewer #1: Yes

3. Have the authors made all data underlying the findings in their manuscript fully available (please refer to the Data Availability Statement at the start of the manuscript PDF file)?

Reviewer #1: Yes

4. Is the manuscript presented in an intelligible fashion and written in standard English?

Reviewer #1: Yes

5. Review Comments to the Author

Reviewer #1: …. Have carried out a study detailing health-seeking behavior of women considering different cultural beliefs. The findings especially of the prevention of women in sleeping under insecticide treated bed-nets during menstruation is very profound and critical in re-orientation of various beliefs system that hinder malaria and other infectious disease control measures.

Major comments:

Introduction:

Comments 1: The authors are still using the epidemiological statistics of malaria from 2021 and 2020 when there is an updated 2023 world malaria report. The authors should use the updated data.

In addition to individuals residing in high, moderate or low endemic districts/regions of Nepal, the authors should provide the prevalence of malaria in Nepal, and specifically Kanchanpur district, Belauri municipal where available.

In the last paragraph of the introduction, the authors stated that “ previous studies have reported……..” but end up providing reference to only one of such studies.

Minor comments:

Plasmodium, Anopheles etc, please italicise the scientific names of different species.

Results: “Large majority of respondents….” The use of large majority at the same time is a tautology.

6. PLOS authors have the option to publish the peer review history of their article (what does this mean?). If published, this will include your full peer review and any attached files.

**Do you want your identity to be public for this peer review?** For information about this choice, including consent withdrawal, please see our Privacy Policy.

Reviewer #1: No

---

## [Decision Letter · Decision Letter 1]

6 Jun 2024

PGPH-D-24-00163R1

Unveiling the significance of women's role in health-seeking behavior during suspected malaria fever in risk populations of Nepal: Mixed methods cross-sectional study

Dear Dr. Paudel,

Thank you for submitting your manuscript to PLOS Global Public Health. After careful consideration, we feel that it has merit but does not fully meet PLOS Global Public Health’s publication criteria as it currently stands. Therefore, we invite you to submit a revised version of the manuscript that addresses the points raised during the review process. Points that still need to be addressed are appended below this message.

We look forward to receiving your revised manuscript.

Kind regards,

Nsa Dada, MSc, PhD

Academic Editor

Journal Requirements:

Additional Editor Comments (if provided):

Dr. Paudel,

Thank you for taking the time to revise and resubmit your manuscript. We have now reviewed the revised manuscript and below are some points that still need to be addressed:

- check and correct malaria incidence stats; also note that malaria incidence has increased since 2020

- in the introduction, provide more background on healthcare seeking behavior, especially within the context of the primary research objective. Currently, there are only four lines of text about healthcare seeking behavior in the introduction, which is insufficient to provide adequate context for the study

- in the discussion section, expand more on the results regarding women's bed net usage during menstruation i.e. what do the results mean especially in the context of the study population and their characteristics? The authors indicate that this is a knowledge gap but do not discuss the outcome of the current study in this regard and how it fills this gap

- conflicting information is provided in the data availability statement. This needs to be resolved

Reviewers' comments:

Reviewer's Responses to Questions

**Comments to the Author**

1. If the authors have adequately addressed your comments raised in a previous round of review and you feel that this manuscript is now acceptable for publication, you may indicate that here to bypass the “Comments to the Author” section, enter your conflict of interest statement in the “Confidential to Editor” section, and submit your "Accept" recommendation.

Reviewer #1: (No Response)

2. Does this manuscript meet PLOS Global Public Health’s publication criteria? Is the manuscript technically sound, and do the data support the conclusions? The manuscript must describe methodologically and ethically rigorous research with conclusions that are appropriately drawn based on the data presented.

Reviewer #1: Yes

3. Has the statistical analysis been performed appropriately and rigorously?

Reviewer #1: Yes

4. Have the authors made all data underlying the findings in their manuscript fully available (please refer to the Data Availability Statement at the start of the manuscript PDF file)?

Reviewer #1: Yes

5. Is the manuscript presented in an intelligible fashion and written in standard English?

Reviewer #1: Yes

6. Review Comments to the Author

Reviewer #1: (No Response)

7. PLOS authors have the option to publish the peer review history of their article (what does this mean?). If published, this will include your full peer review and any attached files.

**Do you want your identity to be public for this peer review?** For information about this choice, including consent withdrawal, please see our Privacy Policy.

Reviewer #1: No

---

## [Decision Letter · Decision Letter 2]

20 Aug 2024

Unveiling the significance of women's role in health-seeking behavior during suspected malaria fever in risk populations of Nepal: Mixed methods cross-sectional study

PGPH-D-24-00163R2

Dear Dr Paudel,

We are pleased to inform you that your manuscript 'Unveiling the significance of women's role in health-seeking behavior during suspected malaria fever in risk populations of Nepal: Mixed methods cross-sectional study' has been provisionally accepted for publication in PLOS Global Public Health.

Best regards,

Julia Robinson

Executive Editor

Reviewer Comments (if any, and for reference):

Reviewer's Responses to Questions

**Comments to the Author**

1. If the authors have adequately addressed your comments raised in a previous round of review and you feel that this manuscript is now acceptable for publication, you may indicate that here to bypass the “Comments to the Author” section, enter your conflict of interest statement in the “Confidential to Editor” section, and submit your "Accept" recommendation.

Reviewer #1: All comments have been addressed

2. Does this manuscript meet PLOS Global Public Health’s publication criteria? Is the manuscript technically sound, and do the data support the conclusions? The manuscript must describe methodologically and ethically rigorous research with conclusions that are appropriately drawn based on the data presented.

Reviewer #1: Yes

3. Has the statistical analysis been performed appropriately and rigorously?

Reviewer #1: Yes

4. Have the authors made all data underlying the findings in their manuscript fully available (please refer to the Data Availability Statement at the start of the manuscript PDF file)?

Reviewer #1: Yes

5. Is the manuscript presented in an intelligible fashion and written in standard English?

Reviewer #1: Yes

6. Review Comments to the Author

Reviewer #1: (No Response)

7. PLOS authors have the option to publish the peer review history of their article (what does this mean?). If published, this will include your full peer review and any attached files.

**Do you want your identity to be public for this peer review?** For information about this choice, including consent withdrawal, please see our Privacy Policy.

Reviewer #1: **Yes: **Oboh Mary Aigbiremo
